# Drosophila gustatory projections are segregated by taste modality and connectivity

**Stefanie Engert[1,2], Gabriella R Sterne[1], Davi D Bock[2†], Kristin Scott[1]\***

[1]University of California, Berkeley, Berkeley, United States; [2]Janelia Research Campus, Howard Hughes Medical Institute, Ashburn, United States

**Abstract** Gustatory sensory neurons detect caloric and harmful compounds in potential food and convey this information to the brain to inform feeding decisions. To examine the signals that gustatory neurons transmit and receive, we reconstructed gustatory axons and their synaptic sites in the adult *Drosophila melanogaster* brain, utilizing a whole-brain electron microscopy volume. We reconstructed 87 gustatory projections from the proboscis labellum in the right hemisphere and 57 from the left, representing the majority of labellar gustatory axons. Gustatory neurons contain a nearly equal number of interspersed pre- and postsynaptic sites, with extensive synaptic connectivity among gustatory axons. Morphology- and connectivity-based clustering revealed six distinct groups, likely representing neurons recognizing different taste modalities. The vast majority of synaptic connections are between neurons of the same group. This study resolves the anatomy of labellar gustatory projections, reveals that gustatory projections are segregated based on taste modality, and uncovers synaptic connections that may alter the transmission of gustatory signals.

**\*For correspondence:**
kscott@berkeley.edu

**Present address:** †University of Vermont, Burlington, United States

**Competing interest:** The authors declare that no competing interests exist.

## Editor's evaluation

The authors reconstructed the axons of gustatory receptor neurons from the labellum in an EM volume of a whole adult *Drosophila* brain. The authors were able to correlate the EM data with light microscopic data in terms of the identity of neurons reconstructed, thus enabling the use of published functional data already available in terms of different taste modalities. This revealed that extensive synaptic connections are found between neurons of the same modality. This article will be of interest to neuroscientists working in the field of circuits and behavior, especially feeding behavior.

## Introduction

All animals have specialized sensory neurons dedicated to the detection of the rich variety of chemicals in the environment that indicate the presence of food sources, predators, and conspecifics. Gustatory sensory neurons have evolved to detect food-associated chemicals and report the presence of caloric or potentially harmful compounds. Examining the activation and modulation of gustatory sensory neurons is essential as it places fundamental limits on the taste information that is funneled to the brain and integrated to form feeding decisions.

The *Drosophila melanogaster* gustatory system is an attractive model to examine the synaptic transmission of gustatory neurons. Molecular genetic approaches coupled with physiology and behavior have established five different classes of gustatory receptor neurons (GRNs) in adult *Drosophila* that detect different taste modalities. One class, expressing members of the gustatory receptor (GR) family, including Gr5a and Gr64f, detects sugars and elicits acceptance behavior (*Dahanukar et al.,*

2001; *Dahanukar et al., 2007*; *Thorne et al., 2004*; *Wang et al., 2004*). A second class expressing different GRs, including Gr66a, detects bitter compounds and mediates rejection behavior (*Thorne et al., 2004*; *Wang et al., 2004*; *Weiss et al., 2011*). A third class contains the ion channel Ppk28 and detects water (*Cameron et al., 2010*; *Chen et al., 2010*). The fourth expresses the Ir94e ionotropic receptor, whereas the fifth contains the Ppk23 ion channel (*Jaeger et al., 2018*; *Thistle et al., 2012*). These cells have been proposed to mediate detection of low-salt and high-salt concentrations, respectively (*Jaeger et al., 2018*). In addition to well-characterized gustatory neurons and a peripheral strategy for taste detection akin to mammals (*Yarmolinsky et al., 2009*), the reduced number of neurons in the *Drosophila* nervous system and the availability of electron microscopy (EM) brain volumes offer the opportunity to examine gustatory transmission with high resolution.

The cell bodies of gustatory neurons are housed in sensilla on the body surface, including the proboscis labellum, an external mouthparts organ that detects taste compounds prior to ingestion (*Stocker, 1994*). Gustatory neurons from each labellum half send bilaterally symmetric axonal projections to the subesophageal zone (SEZ) of the fly brain via the labial nerves. Gustatory axons terminate in the medial SEZ in a region called the anterior central sensory center (ACSC) (*Hartenstein et al., 2018*; *Miyazaki and Ito, 2010*; *Thorne et al., 2004*; *Wang et al., 2004*). Axons from bitter gustatory neurons send branches to the midline and form an interconnected medial ring, whereas other gustatory axons remain ipsilateral and anterolateral to bitter projections. Although projections of different gustatory classes have been mapped using light-level microscopy, the synaptic connectivity of gustatory axons in adult *Drosophila* is largely unexamined.

To explore the connectivity of GRNs and lay the groundwork to study gustatory circuits with synaptic resolution, we used the recently available Full Adult Fly Brain (FAFB) EM dataset (*Zheng et al., 2018*) to fully reconstruct gustatory axons and their synaptic sites. We reconstructed 87 GRN axonal projections in the right hemisphere and 57 in the left, representing between 83–96% and 54–63% of the total expected, respectively (*Jaeger et al., 2018*; *Stocker, 1994*). By annotating chemical synapses, we observed that GRNs contain a nearly equal number of interspersed pre- and postsynaptic sites. Interestingly, GRNs synapse onto and receive synaptic inputs from many other GRNs. Using morphology- and connectivity-based clustering, we identified six distinct neural groups, likely representing groups of GRNs that recognize different taste modalities. Our study reveals extensive anatomical connectivity between GRNs within a taste modality, arguing for presynaptic processing of taste information prior to transmission to downstream circuits.

## Results

### GRN axons contain presynaptic and postsynaptic sites

To systematically characterize gustatory inputs and outputs, we traced gustatory axons in the FAFB volume (*Zheng et al., 2018*). Tracing was performed manually using the annotation platform CATMAID (*Saalfeld et al., 2009*). The GRNs from the proboscis labellum send axons through the labial nerve to the SEZ (*Figure 1A*). The labial nerve is a compound nerve, carrying sensory axons from the labellum, maxillary palp, and eye, as well as motor axons innervating proboscis musculature (*Hampel et al., 2017*; *Hartenstein et al., 2018*; *Miyazaki and Ito, 2010*; *Nayak and Singh, 1983*; *Rajashekhar and Singh, 1994*). Different sensory afferents occupy different domains in the SEZ, with labellar gustatory axons terminating in the ACSC (*Hartenstein et al., 2018*; *Miyazaki and Ito, 2010*; *Thorne et al., 2004*; *Wang et al., 2004*; *Figure 1A*). Therefore, to trace gustatory axons, we began by tracing neurites in the right labial nerve, readily identifiable in the EM dataset (*Figure 1B and C*), and selected fibers that terminated in the anterior central SEZ to trace synaptic completion (*Zheng et al., 2018*).

In tracing axons, we found that neurites with small- to medium-sized diameters in the dorsomedial labial nerve (*Figure 1C*) projected along a single neural tract (*Figure 1D*) to the anterior central region of the SEZ. This neural tract served as an additional site to select arbors for reconstruction. Individual fibers followed along the same tract and showed variation in terminal branching (*Figure 1E*). In total, we identified 87 axonal projections in the right hemisphere. Tracing from the left labial nerve and neural tract in the left hemisphere, we identified 57 additional projections. Misalignments in the EM volume precluded identification of additional GRNs in the left hemisphere. Because there are 90–104 GRNs per labellum (*Jaeger et al., 2018*; *Stocker, 1994*), we estimate that we have identified 83–96% of the GRN fibers from the right labellum and 54–63% from the left. The projections from

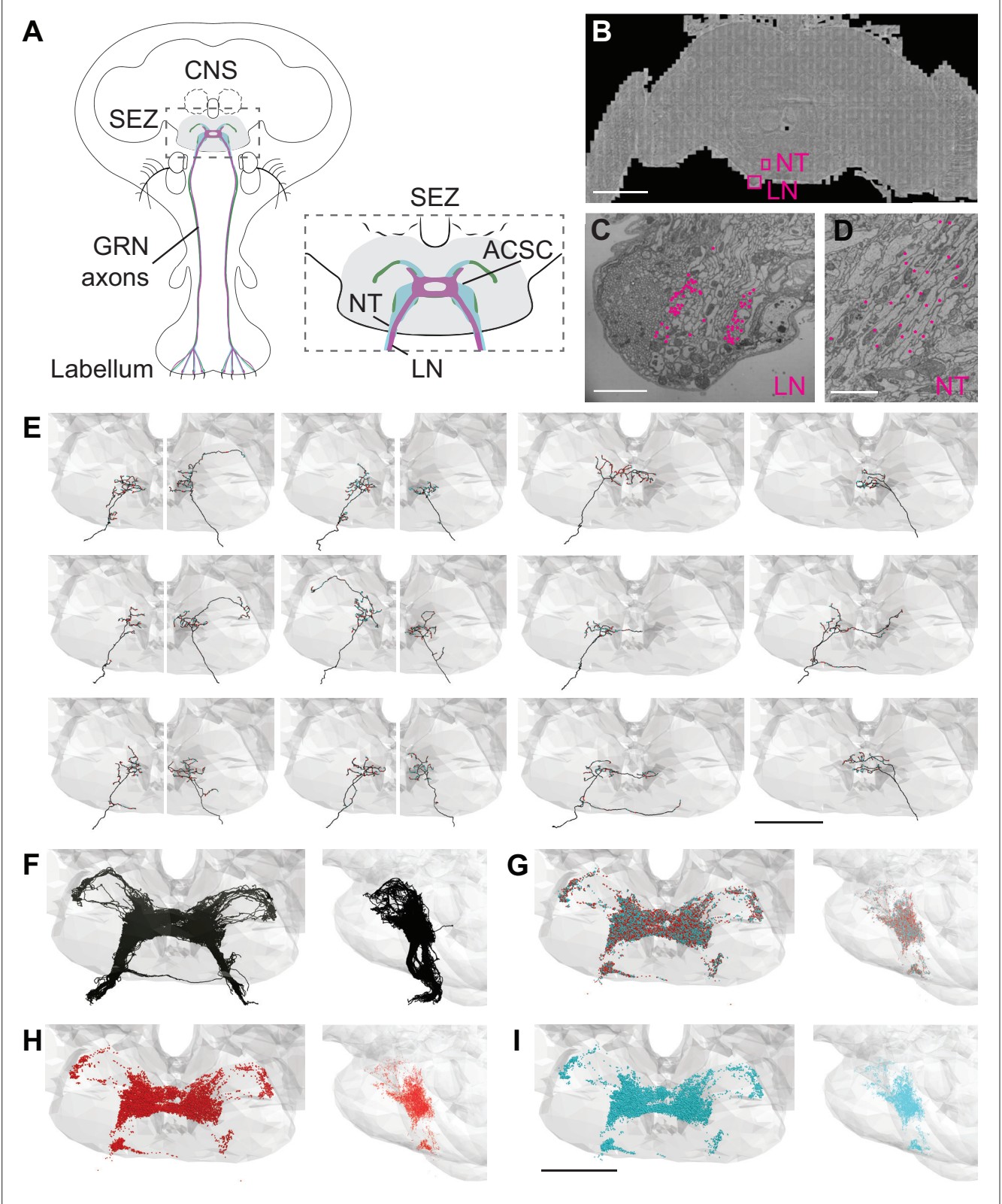

**Figure 1.** Electron microscopy (EM)-based reconstructions of gustatory receptor neurons (GRNs) and synaptic sites. (**A**) Schematic showing GRNs in the proboscis labellum and their axons terminating in the subesophageal zone (SEZ) (gray) in the central nervous system (CNS) (left). Close-up of SEZ (boxed region on left) (gray), noting the labial nerve (LN) and GRN neural tract (NT). GRNs that detect bitter (magenta), sugar (green), and water (blue) terminate in the anterior central sensory center (ACSC) region of the SEZ. (**B**) Location of the LN and NT containing GRNs of the right hemisphere in

*Figure 1 continued on next page*

*Figure 1 continued*

the FAFB dataset (Z slice 3320, scale bar = 100 µM). (**C**) Cross-section of the labial nerve with traced GRNs indicated by asterisks (Z slice 3320, scale bar = 5 µM). (**D**) Neural tract with traced GRNs indicated by asterisks (Z slice 2770, scale bar = 5 µM). (**E**) Examples of reconstructed GRNs with presynaptic (red) and postsynaptic (blue) sites, scale bar = 50 µM. (**F–I**) Frontal and sagittal views of all reconstructed GRN axons (**F**), all presynaptic (red) and postsynaptic (blue) sites (**G**), presynaptic sites alone (**H**), and postsynaptic sites alone (**I**) Scale bar = 50 µM.

The online version of this article includes the following figure supplement(s) for figure 1:

**Figure supplement 1.** Morphology and connectivity of reconstructed gustatory receptor neuron (GRN) skeletons.

the left and right labial nerves are symmetric and converge in a dense web in the anterior central SEZ (*Figure 1F*). This arborization pattern recapitulates the labellar sensory projections of the ACSC (*Hartenstein et al., 2018*). We confirmed that the reconstructed neurites overlap with the known projection pattern of sugar and bitter GRNs in the registered fly brain template (*Figure 1—figure supplement 1*; *Bogovic et al., 2020*), demonstrating that we have identified and traced GRNs.

In addition to the skeleton reconstructions, we manually annotated pre- and postsynaptic sites. The presence of T-shaped structures characteristic of presynaptic release sites ('T bars'), synaptic vesicles, and a synaptic cleft was used to identify a synapse, consistent with previous studies (*Zheng et al., 2018*). Synapses are sparse along the main neuronal tract and abundant at the terminal arborizations (*Figure 1E*). Each GRN has a large number of pre- and postsynaptic sites intermixed along the arbors (*Figure 1E and G–I*), characteristic of fly neurites (*Bates et al., 2020a*; *Meinertzhagen, 2018*; *Olsen and Wilson, 2008*; *Takemura et al., 2017*). On average, a GRN contains 175 (±6 SE) presynaptic sites and 168 (±6 SE) postsynaptic sites, with individual GRNs showing wide variation in pre- and post-synapse number (*Figure 1—figure supplement 1B*). GRNs are both pre- and postsynaptic to other GRNs, with each GRN receiving between 2% and 66% (average = 39%) of its synaptic input from other GRNs (*Figure 1—figure supplement 1C*). The large number of synapses between GRNs suggests that communication between sensory neurons may directly regulate sensory output.

## Different GRN classes can be identified by morphology and connectivity

*Drosophila* GRNs comprise genetically defined, discrete populations that are specialized for the detection of specific taste modalities (*Wang et al., 2004*; *Cameron et al., 2010*; *Jaeger et al., 2018*). As the EM dataset does not contain molecular markers to distinguish between GRNs recognizing different taste modalities, we set out to identify subpopulations of reconstructed GRNs based on their anatomy and connectivity.

We performed hierarchical clustering of GRN axons to define different subpopulations based on their morphology and synaptic connectivity. GRNs of the right hemisphere were used in this analysis as the dataset is more complete. Each traced skeleton was registered to a standard template brain (*Bogovic et al., 2020*), and morphological similarity was compared pairwise using NBLAST in an all-by-all matrix (*Costa et al., 2016*). Then, GRN-GRN connectivity was added for each GRN skeleton and the resulting merged matrix was min/max scaled. We then used Ward's method to hierarchically cluster GRNs into groups (Ward 1963). We chose six groups as the number that minimizes within-cluster variance (*Figure 2—figure supplement 1A*; *Braun et al., 2010*). Each group is composed of 7–23 GRNs that occupy discrete zones in the SEZ and share anatomically similar terminal branches (*Figure 2*).

To evaluate whether the different groups represent GRNs detecting different taste modalities, we compared the anatomy of each group in the right hemisphere with that of known GRN classes using mean NBLAST scores. We registered EM reconstructed GRN projections and GRN projections from immunostained brains to the same standard brain template for direct comparisons (*Bogovic et al., 2020*). For each group, we performed pairwise NBLAST comparisons with bitter (Gr66a; *Wang et al., 2004*; *Thorne et al., 2004*), sugar (Gr64f; *Dahanukar et al., 2007*), water (Ppk28; *Cameron et al., 2010*; *Chen et al., 2010*), and candidate low-salt (Ir94e; *Croset et al., 2016*; *Jaeger et al., 2018*) GRN projections. There is not a specific genetic marker for candidate high-salt projections as Ppk23 labels both bitter and high-salt GRNs (*Jaeger et al., 2018*). These comparisons (see section 'NBLAST analysis for taste modality assignment') yielded a GRN category best match for each group, illustrated by overlays in the three-dimensional standard fly brain template (*Figure 3*). Groups 1 and 2 best match

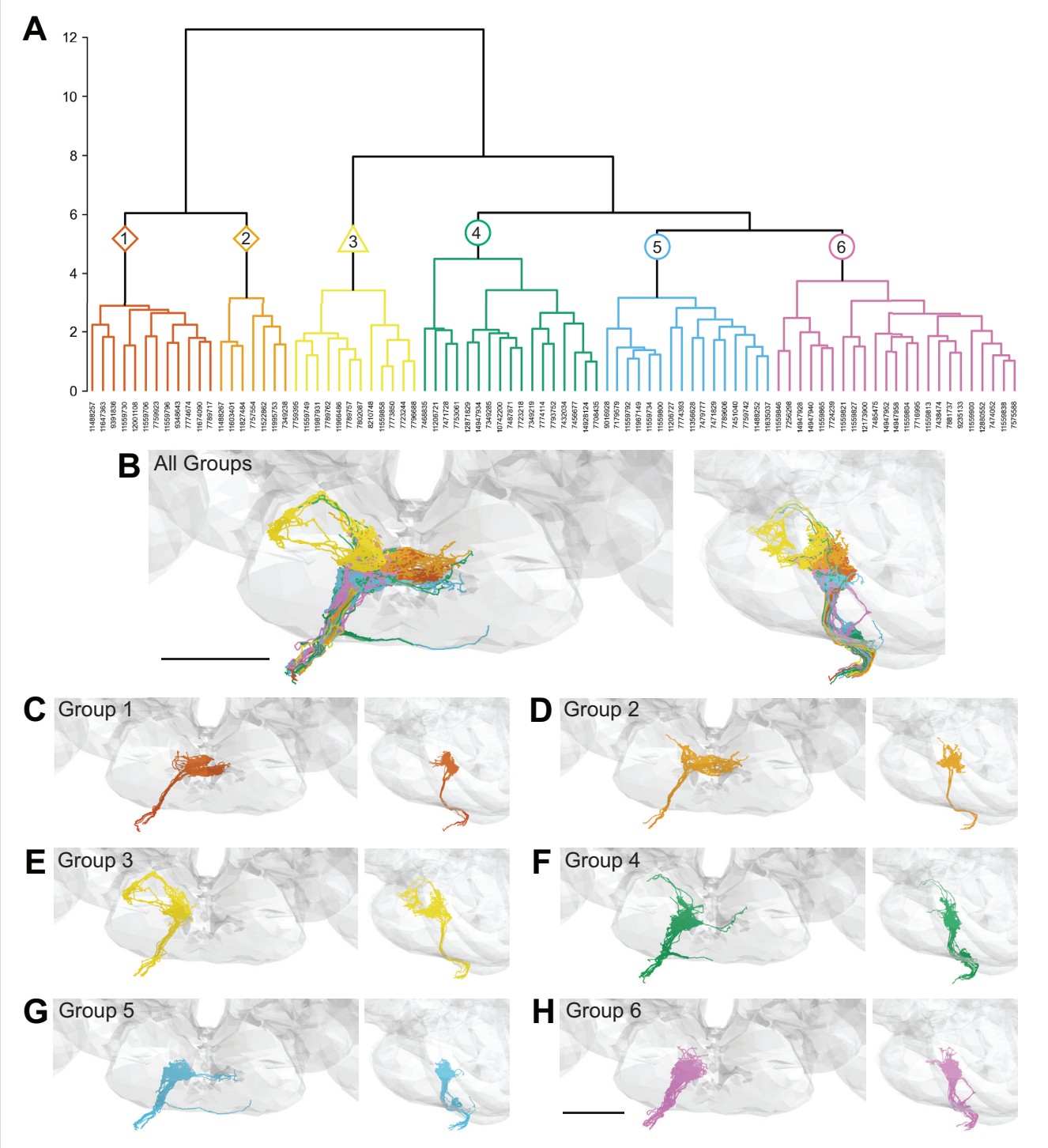

**Figure 2.** Morphology- and connectivity-based clustering generates distinct groups of gustatory receptor neurons (GRNs). (**A**) Tree denoting relative similarity of GRNs based on morphology and connectivity of GRNs in the right hemisphere. (**B**) Frontal and sagittal views of all GRN groups, colored according to (**A**). (**C–H**) Frontal and sagittal views of group 1–6 GRNs, scale bar = 50 μM.

The online version of this article includes the following figure supplement(s) for figure 2:

**Figure supplement 1.** Ward's joining cost and the differential of Ward's joining cost for hierarchical clustering of gustatory receptor neurons (GRNs) in the right hemisphere.

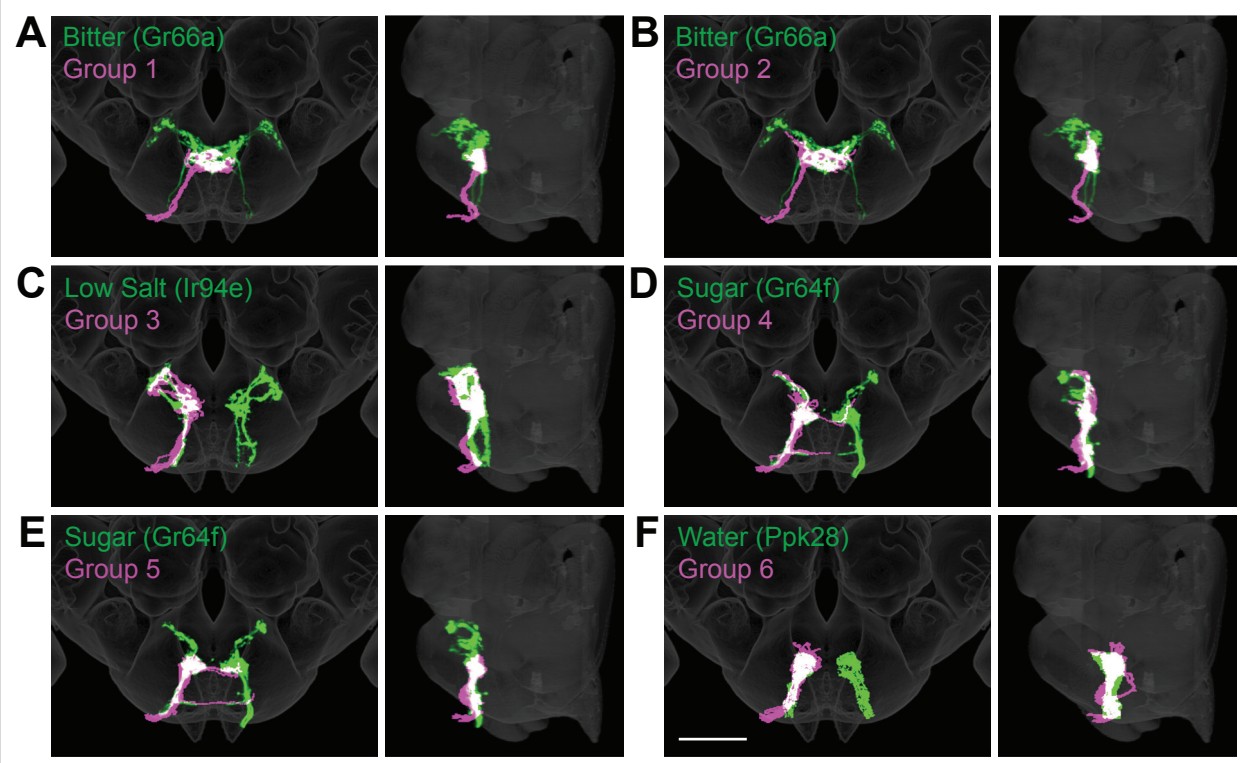

**Figure 3.** Anatomy of different gustatory receptor neuron (GRN) groups overlays with GRNs of different taste categories. NBLAST comparisons yielded best matches of electron microscopy (EM) groups and GRNs of different taste classes. (**A–F**) Overlain are EM groups 1–6 (magenta) and best NBLAST match of GRN class (immunohistochemistry, green), frontal view (left), and sagittal view (right), scale bar = 50 μM.

The online version of this article includes the following figure supplement(s) for figure 3:

**Figure supplement 1.** Morphology- and connectivity-based clustering generates distinct groups of gustatory receptor neurons (GRNs).

**Figure supplement 2.** Ward's joining cost and the differential of Ward's joining cost for hierarchical clustering of gustatory receptor neurons (GRNs) in the left hemisphere.

bitter projections, forming a characteristic medial ringed web (*Figure 3A and B*). Group 3 projections show greatest similarity to low-salt GRNs, with distinctive dorsolateral branches (*Figure 3C*). Groups 4–6 are anatomically very similar, and identity assignments are tentative. Groups 4 and 5 best match sugar GRNs (*Figure 3D and E*). Because group 4 contains a dorsolateral branch seen in Gr64f projections and not seen in group 5 projections, we hypothesize that group 4 is composed of sugar GRNs and that the remaining group 5 is composed of high-salt GRNs. Group 6 best matches water GRNs (*Figure 3F*). Thus, morphological and connectivity clustering suggests molecular and functional identities of different GRNs.

An identical clustering analysis of GRNs from the left hemisphere yielded seven groups of 4–15 neurons (*Figure 3—figure supplements 1–2*). Groups 1 and 2 best match bitter projections and group 6 best matches low-salt projections (see section 'NBLAST analysis for taste modality assignment'), with anatomy consistent with known projection patterns. Other groups are not well-resolved (see section 'NBLAST analysis for taste modality assignment'), arguing that a more complete dataset is necessary to resolve GRN categories in the left hemisphere.

## GRNs are highly interconnected via chemical synapses

As GRNs have a large number of synaptic connections with other GRNs (*Figure 1—figure supplement 1C*), we examined whether synapses exist exclusively between neurons of the same group, likely representing the same taste modality, or between multiple groups. The all-by-all connectivity matrix illustrated blocks of connectivity within groups, with fewer connections between groups (*Figure 4A*). To quantify this, we summed all GRN-GRN connections within and between groups. This analysis revealed that most synapses are between neurons of the same group (79%), while only 21% of the

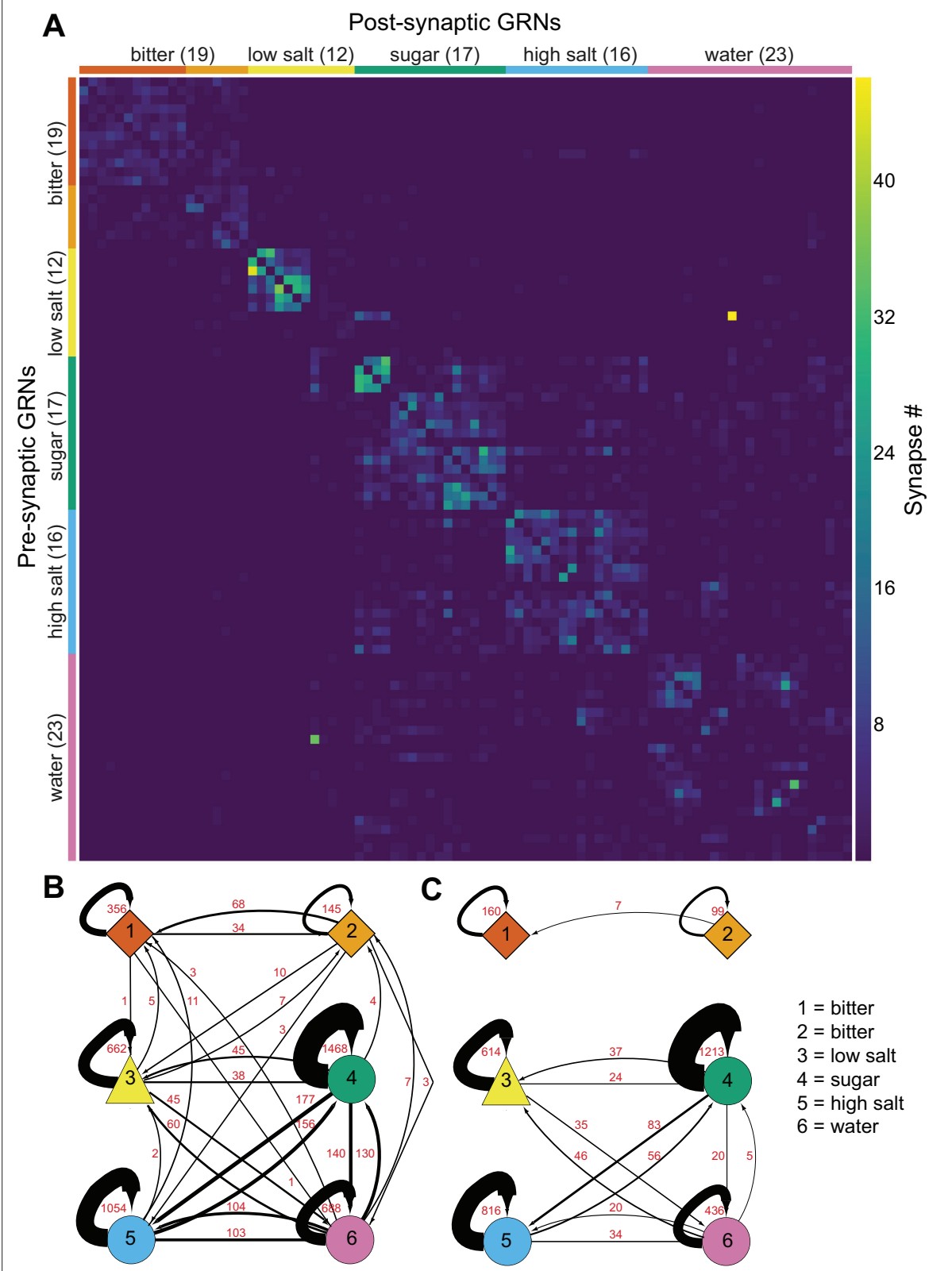

**Figure 4.** Gustatory receptor neurons (GRNs) are highly interconnected via chemical synapses. (**A**) Connectivity matrix of GRNs in the right hemisphere. GRN groups are color-coded and ordered according to **Figure 2**, with number of GRNs/group in parentheses. Color coding within the matrix indicates the number of synapses from the pre- to the postsynaptic neuron, indicated in the legend. (**B**) Connectivity between GRN groups. Colors correspond to

*Figure 4 continued*

groups in *Figure 2*. Arrow thickness scales with the number of synapses, indicated in red. (**C**) Connectivity between GRN groups as in (**B**), showing only connections of five or more synapses. Group # and corresponding taste category are noted on the right.

The online version of this article includes the following figure supplement(s) for figure 4:

**Figure supplement 1.** Predicted neurotransmitters expressed by gustatory receptor neurons (GRNs) of the right hemisphere.

**Figure supplement 2.** Predicted neurotransmitters expressed by gustatory receptor neurons (GRNs) of the left hemisphere.

synapses are between GRNs of different groups (*Figure 4B*). For example, group 4 neurons receive 1468 synapses from other group 4 neurons and 38 from group 3, 156 from group 5, and 130 from group 6 neurons. Focusing on connections of five or more synapses between GRN pairs, representing high-confidence connections (*Buhmann et al., 2021*; *Li et al., 2020*; *Takemura et al., 2013*; *Takemura et al., 2015*), resulted in the elimination of some but not all between-group connections (*Figure 4C*), with between-group connections representing only 10% of all GRN connections.

The large numbers of chemical synapses between GRNs within a group may provide a mechanism to amplify signals of the same taste modality. In contrast, weak connectivity between GRNs of different groups may serve to integrate taste information from different modalities before transmission to downstream circuitry. We note that misclassification of individual GRNs in the clustering analysis may result in over- or underestimates of GRN connectivity within and between groups.

Neurotransmitter predictions of GRNs, in general, do not predict a clear majority neurotransmitter (*Figure 4—figure supplements 1 and 2*; *Eckstein et al., 2020*). This suggests that GRNs may release multiple neurotransmitters or that neurotransmitter predictions should be considered uncertain until further testing (*Eckstein et al., 2020*).

## Interactions between sugar and water GRNs are not observed by calcium or voltage imaging

To examine whether the small number of connections between GRNs of different taste modalities results in cross-activation of GRNs detecting different primary tastant classes, we tested if activation of one GRN class results in propagation of activity to other GRN classes in vivo. To test for interactions between GRNs of different taste modalities, we undertook calcium and voltage imaging studies in which we monitored the response of a GRN class upon activation of other GRN classes.

We expressed the calcium indicator GCaMP6s in genetically defined sugar-, water-, or bitter-sensitive GRNs to monitor excitatory responses upon artificial activation of different GRN classes. To ensure robust and specific activation of GRNs, we expressed the mammalian ATP receptor P2X2 in sugar, water, or bitter GRNs, and activated the GRNs with an ATP solution presented to the fly proboscis while imaging gustatory projections in the brain (*Yao et al., 2012*; *Harris et al., 2015*). Expressing both P2X2 and GCaMP6s in sugar, water, or bitter GRNs elicited strong excitation upon ATP presentation (*Figure 5A–B and G–H*, *Figure 5—figure supplement 1C and D*, *Figure 5—figure supplement 2C and D*, *Figure 5—figure supplement 3C and D*), demonstrating the effectiveness of this method. Activation of sugar or water GRNs did not activate bitter cells, nor did bitter cell activation elicit responses in sugar or water axons (*Figure 5—figure supplement 1E–H*, *Figure 5—figure supplement 2E and F*, *Figure 5—figure supplement 3G and H*). Similarly, we did not observe responses in sugar GRNs upon water GRN activation (*Figure 5C and D*, *Figure 5—figure supplement 2I and J*) or responses in water GRNs upon sugar GRN activation (*Figure 5I and J*, *Figure 5—figure supplement 3E and F*). To examine whether interactions between modalities are modulated by the feeding state of the fly, we performed the activation and imaging experiments in both fed and starved flies (*Figure 5—figure supplements 1–6*). These experiments did not reveal feeding state-dependent interactions between GRN populations. To examine whether inhibitory interactions might exist between two GRN classes, we expressed the voltage indicator ArcLight (*Cao et al., 2013*), which reliably reports hyperpolarization, in sugar GRNs while activating water GRNs via P2X2 and vice versa. These experiments revealed no change in voltage in one appetitive gustatory class upon activation of the other (*Figure 5E–F and K–L*, *Figure 5—figure supplement 7*). Overall, despite the potential for crosstalk between different modalities revealed by EM, we observed no communication between appetitive GRNs by calcium or voltage imaging of gustatory axons.

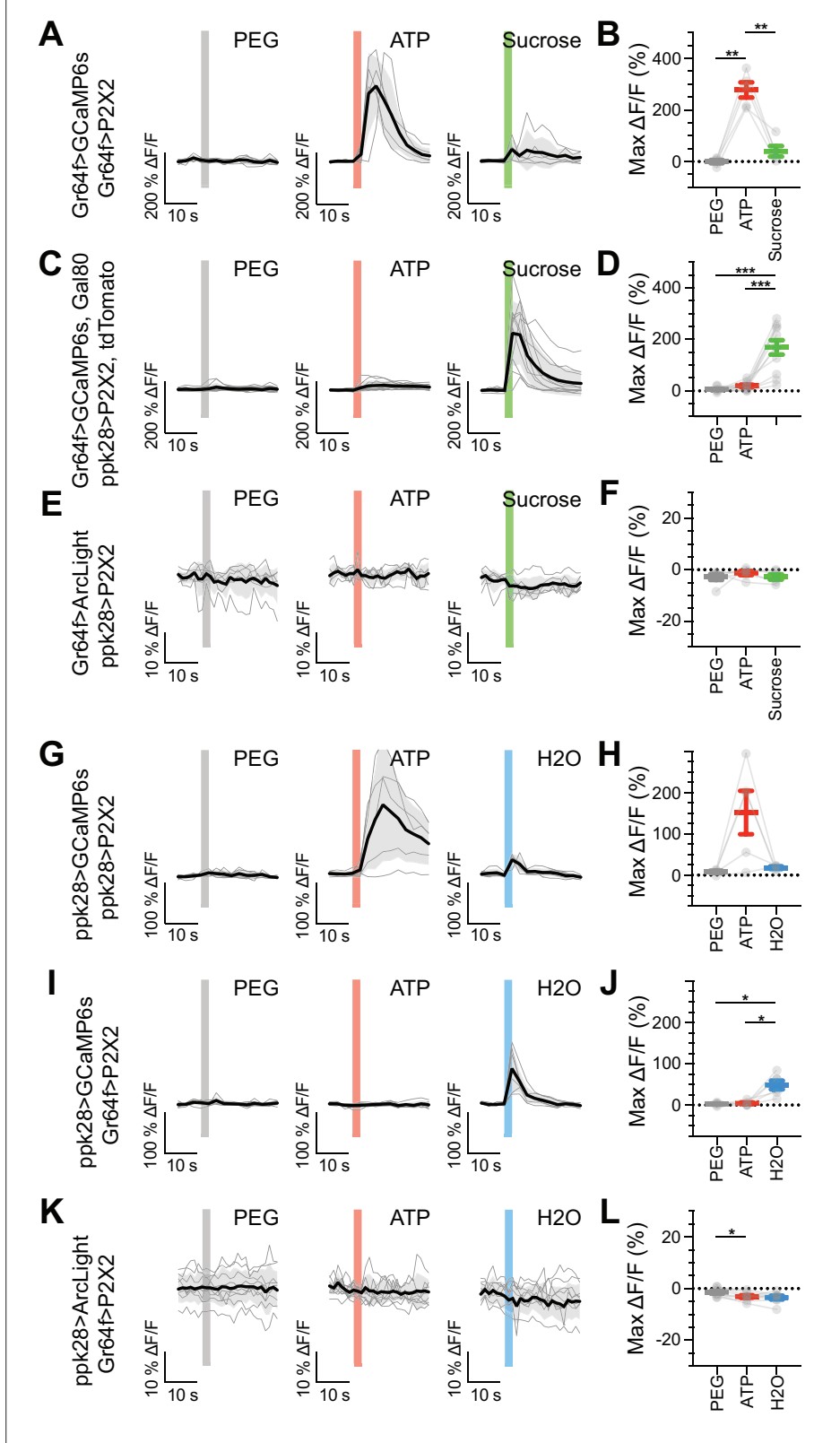

**Figure 5.** Sugar and water gustatory receptor neurons (GRNs) do not activate each other. (**A, B**) Calcium responses of sugar GRNs expressing P2X2 and GCaMP6s to proboscis presentation of PEG as a negative control, ATP to activate P2X2, or sucrose as a positive control. GCaMP6s fluorescence traces (ΔF/F) (**A**) and maximum ΔF/F post-stimulus presentation (**B**), n = 5. Sugar GRNs responded to ATP, but the response to subsequent sucrose

*Figure 5 continued on next page*

*Figure 5 continued*

presentation was attenuated. (**C, D**) GCaMP6s responses of sugar GRNs in flies expressing P2X2 in water GRNs to PEG, ATP, and sucrose delivery, ΔF/F traces (**C**), and maximum ΔF/F graph (**D**), n = 11. (**E, F**) ArcLight responses of sugar GRNs in flies expressing P2X2 in water GRNs, ΔF/F traces (**E**), and maximum ΔF/F graph (**F**), n = 6. (**G, H**) Calcium responses of water GRNs expressing P2X2 and GCaMP6s to proboscis delivery of PEG (negative control), ATP, and water (positive control), ΔF/F traces (**G**), and maximum ΔF/F graph (**H**), n = 5. Water GRNs responded to ATP presentation, but the subsequent response to water was diminished. (**I, J**) GCaMP6s responses of water GRNs in flies expressing P2X2 in sugar GRNs to PEG, ATP, and water, ΔF/F traces (**I**), and maximum ΔF/F graph (**J**), n = 6. (**K, L**) ArcLight responses of water GRNs in flies expressing P2X2 in sugar GRNs to PEG, ATP, and water, ΔF/F traces (**K**), and maximum ΔF/F graph (**L**), n = 9. For all traces, stimulus presentation is indicated by shaded bars. Traces of individual flies to the first of three taste stimulations (shown in *Figure 5—figure supplement 2*, *Figure 5—figure supplement 3*, and *Figure 5—figure supplement 7*) are shown in gray, the average in black, with the SEM indicated by the gray shaded area. Repeated-measures ANOVA with Tukey's multiple-comparisons test, *p<0.05, **p<0.01, ***p<0.001.

The online version of this article includes the following figure supplement(s) for figure 5:

**Figure supplement 1.** Bitter gustatory receptor neurons (GRNs) do not respond to the activation of other GRN classes in fed flies.

**Figure supplement 2.** Sugar gustatory receptor neurons (GRNs) do not respond to the activation of other GRN classes in fed flies.

**Figure supplement 3.** Water gustatory receptor neurons (GRNs) do not respond to the activation of other GRN classes in fed flies.

**Figure supplement 4.** Bitter gustatory receptor neurons (GRNs) do not respond to the activation of other GRN classes in food-deprived flies.

**Figure supplement 5.** Sugar gustatory receptor neurons (GRNs) do not respond to the activation of other GRN classes in food-deprived flies.

**Figure supplement 6.** Water gustatory receptor neurons (GRNs) do not respond to the activation of other GRN classes in food-deprived flies.

**Figure supplement 7.** Sugar and water gustatory receptor neurons (GRNs) do not show voltage responses upon reciprocal activation.

## Discussion

In this study, we characterized different classes of gustatory projections and their interconnectivity by high-resolution EM reconstruction. We identified different projection patterns corresponding to gustatory neurons recognizing different taste modalities. The extensive connections between GRNs of the same taste modality provide anatomical evidence of presynaptic processing of gustatory information.

An emerging theme stemming from EM reconstructions of *Drosophila* sensory systems is that sensory neurons of the same subclass are synaptically connected. In general, different sensory neuron subclasses have spatially segregated axonal termini in the brain, thereby constraining the potential for connectivity. In the adult olfactory system, approximately 40% of the input onto olfactory receptor neurons (ORNs) comes from other ORNs projecting to the same olfactory glomerulus (*Horne et al., 2018*; *Schlegel et al., 2021*; *Tobin et al., 2017*). Similarly, mechanosensory projections from Johnston's organ of the same submodality are anatomically segregated and synaptically connected (*Hampel et al., 2020*). In *Drosophila* larvae, 25% of gustatory neuron inputs are from other GRNs, although functional classes were not resolved (*Miroschnikow et al., 2018*). In the adult *Drosophila* gustatory system, we also find that GRNs are interconnected, with approximately 39% of GRN input coming from other GRNs. Consistent with other classes of sensory projections, we find that gustatory projections are largely segregated based on taste modality and form connected groups. A general function of sensory–sensory connections seen across sensory modalities may be to enhance weak signals or increase dynamic range.

By clustering neurons based on anatomy and connectivity, we were able to resolve different GRN categories. The distinct morphologies of bitter neurons and candidate low-salt-sensing neurons, known from immunohistochemistry, are recapitulated in the projection patterns of GRN groups 1–3 of the right hemisphere, enabling high-confidence identification. The projections of high-salt-, sugar-, and water-sensing neurons are ipsilateral, with similarities in their terminal arborizations (*Jaeger et al., 2018*; *Wang et al., 2004*). Nevertheless, comparisons between EM and light-level projections argue

that these taste categories are also resolved into different, identifiable clusters. We identified GRN categories as low salt (Ir94e) and high salt (the remaining category) based on previous studies (*Jaeger et al., 2018*) but note that the full complement of tastes that these GRNs detect requires additional investigation. The GRN categories that we identify here are based on anatomical comparisons alone and remain tentative until further examination of taste response profiles of connected second-order neurons, which may now be identified by examining connectivity downstream of GRNs.

Here, we reconstructed 83–96% of the GRNs on the right hemisphere and 54–63% on the left, based on total GRN counts from previous studies (*Jaeger et al., 2018*; *Stocker, 1994*). GRN categories may be further refined upon reconstruction of the entire GRN population or upon analysis that includes postsynaptic partners. In addition, GRNs are found at different locations on the proboscis labellum and are housed in three taste bristle types (*Stocker, 1994*). Segregation based on labellar location or bristle type may further divide the GRN categories described here. Interestingly, in our clustering analysis, we find that bitter projections cluster into two distinct groups. We hypothesize that these different subsets are comprised of bitter GRNs from different taste bristle classes or bitter GRNs with different response properties (*Dweck and Carlson, 2020*).

Examining GRN-GRN connectivity revealed connectivity between GRNs of the same group as well as different groups. While it is tempting to speculate that interactions between different taste modalities may amplify or filter activation of feeding circuits, we were unable to identify cross-activation between sugar and water GRNs by calcium or voltage imaging. It is possible that these interactions are dependent on a feeding state or act on a time frame not examined in this study. Alternatively, activation may fall below the detection threshold of calcium or voltage imaging. Additionally, far fewer synapses occur between anatomical classes than within classes, especially restricting analyses to neurons connected by five or more synapses (*Figure 4C*), suggesting that the few synapses may not be relevant for taste processing. Finally, the anatomy and connectivity-based clustering may not categorize all individual GRNs correctly, and misclassification of GRNs would impact connectivity analyses. Regardless, our studies suggest that presynaptic connectivity between different GRN classes does not substantially contribute to taste processing.

Overall, this study resolves the majority of labellar gustatory projections and their synaptic connections, revealing that gustatory projections are segregated based on taste modality and sensory–sensory connectivity. The identification of GRNs detecting different taste modalities now provides an inroad to enable the examination of the downstream circuits that integrate taste information and guide feeding decisions.

## Materials and methods

**Key resources table**

| Reagent type (species) or resource | Designation | Source or reference | Identifiers | Additional information |
|---|---|---|---|---|
| Genetic reagent (*Drosophila melanogaster*) | Gr64f-Gal4 (II) | *Kwon et al., 2011* | BDSC:57669; FLYB:FBti0162679 | |
| Genetic reagent (*D. melanogaster*) | Gr64f-Gal4 (III) | *Kwon et al., 2011* | BDSC:57668; FLYB: FBti0162678 | |
| Genetic reagent (*D. melanogaster*) | Gr64f-LexA (III) | *Miyamoto et al., 2012* | | |
| Genetic reagent (*D. melanogaster*) | Gr66a-Gal4 (II) | *Scott et al., 2001* | | |
| Genetic reagent (*D. melanogaster*) | Gr66a-LexA (III) | *Thistle et al., 2012* | | |
| Genetic reagent (*D. melanogaster*) | Ppk28-Gal4 (II) | *Cameron et al., 2010* | | |
| Genetic reagent (*D. melanogaster*) | Ppk28-LexA (III) | *Thistle et al., 2012* | | |

*Continued on next page*

*Continued*

| Reagent type (species) or resource | Designation | Source or reference | Identifiers | Additional information |
|---|---|---|---|---|
| Genetic reagent (*D. melanogaster*) | *Ir94e-Gal4* (attp2) | *Croset et al., 2016* | BDSC:81246; FLYB:FBti0202323 | |
| Genetic reagent (*D. melanogaster*) | csChrimsonReporter/Optogenetic effector,*20xUAS- csChrimson::mVenus* in attP18 | *Klapoetke et al., 2014* | BDSC:55134; FLYB:FBst0055134 | |
| Genetic reagent (*D. melanogaster*) | *UAS-Syt-HA;;* | *Robinson et al., 2002* | | |
| Genetic reagent (*D. melanogaster*) | *UAS-P2X2* (chr III) | *Lima and Miesenböck, 2005* | BDSC:91222; FLYB:FBst0091222 | |
| Genetic reagent (*D. melanogaster*) | *UAS-ArcLight* (attp2) | *Cao et al., 2013* | BDSC:51056; FLYB:FBst0051056 | |
| Genetic reagent (*D. melanogaster*) | *LexAop-GCaMP6s* (attp5) | *Chen et al., 2013* | BDSC:44589; FLYB:FBst0044589 | |
| Genetic reagent (*D. melanogaster*) | *LexAop-GCaMP6s* (attp1) | *Chen et al., 2013* | BDSC:44588; FLYB:FBst0044588 | |
| Genetic reagent (*D. melanogaster*) | *LexAop-Gal80* (X) | *Thistle et al., 2012* | | |
| Genetic reagent (*D. melanogaster*) | *UAS-CD8::tdTomato* (chr X) | *Thistle et al., 2012* | | |
| Genetic reagent (*D. melanogaster*) | *UAS-CD8::tdTomato* (II) | *Thistle et al., 2012* | | |
| Antibody | Anti-Brp (mouse monoclonal) | DSHB, University of Iowa, USA | DSHB:Cat# nc82; RRID:AB_2314866 | 1/500 |
| Antibody | Anti-GFP (rabbit polyclonal) | Thermo Fisher Scientific | Thermo Fisher Scientific:Cat# A11122; RRID:AB_221569 | 1/1000 |
| Antibody | Anti-GFP (chicken polyclonal) | Thermo Fisher Scientific | Thermo Fisher Scientific:Cat# A10262; RRID:AB_2534023 | 1/1000 |
| Antibody | Anti-dsRed (rabbit polyclonal) | Takara Bio | Takara Bio:Cat# 632496; RRID: AB_10013483 | 1/1000 |
| Antibody | Anti-rabbit Alexa Fluor 488 (goat polyclonal) | Thermo Fisher Scientific | Thermo Fisher Scientific:Cat# A11034; RRID:AB_2576217 | 1/100 |
| Antibody | Anti-chicken Alexa Fluor 488 (goat polyclonal) | Thermo Fisher Scientific | Thermo Fisher Scientific:Cat# A11039; RRID:AB_2534096 | 1/100 |
| Antibody | Anti-rabbit Alexa Fluor 568 (goat polyclonal) | Thermo Fisher Scientific | Thermo Fisher Scientific:Cat# A11036; RRID:AB_10563566 | 1/100 |
| Antibody | Anti-mouse Alexa Fluor 647 (goat polyclonal) | Thermo Fisher Scientific | Thermo Fisher Scientific:Cat# A21236; RRID:AB_2535805 | 1/100 |
| Chemical compound, drug | Denatonium benzoate | MilliporeSigma | MilliporeSigma:Cat# D5765; CAS:3734-33-6 | |
| Chemical compound, drug | Caffeine | MilliporeSigma | MilliporeSigma:Cat# C53; CAS:58-08-2 | |
| Chemical compound, drug | Sucrose | Thermo Fisher Scientific | Thermo Fisher Scientific:Cat# AAA1558336; CAS:57-50-1 | |

*Continued on next page*

*Continued*

| Reagent type (species) or resource | Designation | Source or reference | Identifiers | Additional information |
|---|---|---|---|---|
| Chemical compound, drug | Polyethylene glycol (MW 3350) | MilliporeSigma | MilliporeSigma:Cat# P4338; CAS:25322-68-3 | |
| Chemical compound, drug | All-trans-retinal | MilliporeSigma | MilliporeSigma:Cat# R2500; CAS:116-31-4 | |
| Software, algorithm | Fiji | *Schindelin et al., 2012* | RRID:SCR_002285 | http://fiji.sc/ |
| Software, algorithm | CATMAID | *Schneider-Mizell et al., 2016* | RRID:SCR_006278 | https://catmaid.readthedocs.io/ |
| Software, algorithm | R Project for Statistical Computing | R Development Core Team, 2018 | RRID:SCR_001905 | https://www.r-project.org/ |
| Software, algorithm | NeuroAnatomy Toolbox | *Jefferis and Manton, 2017* | RRID:SCR_017248 | https://github.com/jefferis/nat |
| Software, algorithm | Python | Python Software Foundation | RRID:SCR_008394 | https://www.python.org/ |
| Software, algorithm | Jupyter Notebook | Project Jupyter | RRID:SCR_018315 | https://jupyter.org/ |
| Software, algorithm | Slidebook | Intelligent Imaging Innovations | RRID:SCR_014300 | https://www.intelligent-imaging.com/slidebook |
| Software, algorithm | GraphPad Prism | GraphPad Software | RRID:SCR_002798 | https://www.graphpad.com/ |
| Software, algorithm | Cytoscape | *Shannon et al., 2003* | RRID:SCR_003032 | https://cytoscape.org/ |
| Software, algorithm | Computational Morphometry Toolkit | *Rohlfing and Maurer, 2003* | RRID:SCR_002234 | https://www.nitrc.org/projects/cmtk/ |

## Experimental animals

Experimental animals were maintained on standard agar/molasses/cornmeal medium at 25°C. For imaging experiments requiring food-deprived animals, flies were placed in vials containing wet kimwipes for 23–26 hr prior to the experiment. For behavioral experiments, flies were placed on food supplemented with 400 µM trans-retinal for 24 hr prior to the experiment.

## EM reconstruction

Neuron skeletons were reconstructed in a serial sectioned transmission EM dataset of the whole fly brain (*Zheng et al., 2018*) using the annotation software CATMAID (*Saalfeld et al., 2009*). GRN projections were identified based on their extension into the labial nerve and localization to characteristic neural tracts in the SEZ. Skeletons were traced to completion either entirely manually or using a combination of an automated segmentation (*Li et al., 2019*) and manual tracing as previously described (*Hampel et al., 2020*). Chemical synapses were annotated manually and neurons were traced to synaptic completion using criteria previously described (*Zheng et al., 2018*). Skeletons were reviewed by a second specialist, so that the final reconstruction presents the consensus assessment of at least two specialists. Skeletons were exported from CATMAID as swc files for further analysis, and images of skeletons were exported directly from CATMAID. FAFB neuronal reconstructions will be available from Virtual Fly Brain (https://fafb.catmaid.virtualflybrain.org/).

## Clustering of GRNs

GRNs were hierarchically clustered based on morphology and connectivity using NBLAST and synapse counts. First, GRN skeletons traced in FAFB were registered to the JRC2018U template (*Bogovic et al., 2020*) and compared in an all-by-all fashion with NBLAST (*Costa et al., 2016*). NBLAST analysis was carried out with the natverse toolkit in R (*Bates et al., 2020b*; R Development Core Team, https://www.r-project.org/). The resulting matrix of 'normalized' NBLAST scores was merged with a second matrix containing all-by-all synaptic connectivity counts for the same GRNs. The resulting merged

matrix was min–max normalized such that all values fall within the range of 0 and 1. The merged, normalized matrix was hierarchically clustered using Ward's method (Ward 1963) in Python (Python Software Foundation, https://www.python.org/) with SciPy (*Virtanen et al., 2020*). The number of groups was chosen based on analysis of Ward's joining cost and the differential of Ward's joining cost.

Connectivity data of GRNs was exported from CATMAID for further analysis, and connectivity diagrams were generated using Cytoscape (*Shannon et al., 2003*).

## NBLAST analysis for taste modality assignment

GRN skeletons traced in FAFB were registered to the JRC2018U template and summed in Fiji to create a composite stack of the combined morphologies of all individual GRNs in a given group (as assigned by morphology and connectivity clustering). The morphology of the composite stack for each group was compared to an image library of GRN projection patterns using NBLAST (*Costa et al., 2016*). The image library contained projection patterns of *Gr66a-Gal4*, *Gr64f-Gal4*, *Ir94e-Gal4*, and *Gr64f-Gal4* brains, three per genotype, registered to the JRC2018U template, prepared as described (see section 'Immunohistochemistry'). Group identity was assigned based on the top hit from the image library. Following NBLAST analysis, the anatomy of each group was compared to the projection pattern of its top hit using VVDViewer.

NBLAST of groups in the right hemisphere against known GRN categories yielded the following top GRN matches (mean NBLAST score): group 1, *Gr66a-Gal4* #1 (0.77986); group 2, *Gr66a-Gal4* #1 (0.83017); group 3, Ir94e-GAL4 #2 (0.73743); group 4, *Gr64f-Gal4* #2 (0.80821); group 5, *Gr64f-Gal4* #2 (0.81091); and group 6, *Ppk28-Gal4* #1 (0.80059). NBLAST of groups in the left hemisphere against known GRN categories yielded the following top GRN matches (NBLAST score): group 1, *Gr66a-Gal4* #3 (0.86974); group 2, *Gr66a-Gal4* #3 (0.88230); group 3, *Gr64f-Gal4* #2 (0.85942); group 4, *Gr64f-Gal4* #2 (0.84788); group 5, *Gr64f-Gal4* #2 (0.87164); group 6, *Ir94e-Gal4* #2 (0.79400); and group 7, *Gr64f-Gal4* #2 (0.78896).

## Calcium and voltage imaging preparation

For imaging studies of GRNs, mated females, 10–21 days post eclosion, were dissected as previously described (*Harris et al., 2015*), so that the brain was submerged in artificial hemolymph (AHL) (*Wang et al., 2003*) while the proboscis was kept dry and accessible for taste stimulation. To avoid occlusion of taste projections in the SEZ, the esophagus was cut. The front legs were removed for tastant delivery to the proboscis. AHL osmolality was assessed as previously described (*Jourjine et al., 2016*) and adjusted according to the feeding status of the animal. In fed flies, AHL of ~250 mOsm was used (*Wang et al., 2003*). The AHL used for starved flies was diluted until the osmolality was ~180 mOsm, consistent with measurements of the hemolymph osmolality in food-deprived flies (*Jourjine et al., 2016*).

## Calcium imaging

Calcium transients reported by GCaMP6s and GCaMP7s were imaged on a 3i spinning disk confocal microscope with a piezo drive and a ×20 water immersion objective (NA = 1). For our studies of GRNs, stacks of 14 z-sections, spaced 1.5 µm apart, were captured with a 488 nm laser for 45 consecutive time points with an imaging speed of ~0.3 Hz and an optical zoom of 2.0. For better signal detection, signals were binned 8 × 8, except for Gr64f projections, which underwent 4 × 4 binning.

## Voltage imaging

Voltage responses reported by ArcLight were imaged similarly to the calcium imaging studies. To increase imaging speed, the number of z planes was reduced to 10, and the exposure time was decreased from 100 to 75 ms, resulting in an imaging speed of ~0.7 Hz. To maintain a time course comparable to that of the calcium imaging experiments of GRNs, the number of time points was increased to 90. Signals were binned 8 × 8 in each experiment.

## Taste stimulations

Taste stimuli were delivered to the proboscis via a glass capillary as previously described (*Harris et al., 2015*). For GRN studies, each fly was subjected to three consecutive imaging sessions, each consisting of a taste stimulation at time point 15, 25, and 35 (corresponding to 30, 50.5, 71.5 s). During the first

imaging session, the fly was presented with a tasteless 20% polyethylene glycol (PEG, average molecular weight 3350 g/mol) solution, acting as a negative control. PEG was used in all solutions except water solutions as this PEG concentration inhibits activation of water GRNs (*Cameron et al., 2010*). This was followed in the second session with stimulations with 100 mM ATP in 20%PEG. In the last imaging session, each fly was presented with a tastant acting as a positive control in 20% PEG (Gr64f: 1 M sucrose; Gr66a: 100 mM caffeine, 10 mM denatonium benzoate; ppk28: $H_2O$; ppk23: 1 M KCl in 20% PEG).

### Imaging analysis

Image analysis was performed in Fiji (*Schindelin et al., 2012*). Z stacks for each time point were converted into maximum z-projections for further analysis. After combining these images into an image stack, they were aligned using the StackReg plugin in Fiji to correct for movement in the xy plane (*Thévenaz et al., 1998*).

For our exploration of interactions between GRN subtypes, one region of interest (ROI) was selected encompassing the central arborization of the taste projection in the left or right hemisphere of the SEZ in each fly. Whether the projection in the left or right hemisphere was chosen depended on the strength of their visually gauged response to the positive control. The exception was Gr66a projections, in which the entire central projection served as ROI. If projections did not respond strongly to at least two of the three presentations of the positive control, the fly was excluded from further analysis. If projections responded to two or more presentations of the negative control, the fly was excluded from further analysis. A large ROI containing no GCaMP signal was chosen in the lateral SEZ to determine background fluorescence.

In calcium imaging experiments, the first five time points of each imaging session were discarded, leaving 40 time points for analysis with taste stimulations at time points 10, 20, and 30. The average fluorescence intensity of the background ROI was subtracted at each time point from that of the taste projection ROI. F0 was then defined as the average fluorescence intensity of the taste projection ROI post background subtraction of the first five time points. ΔF/F (%) was calculated as 100% * (F(t) - F0)/F0. Voltage imaging experiments were analyzed similarly, with 10 initial time points discarded for a total of 80 time points in the analysis and tastant presentations at time points 20, 40, and 60.

### Quantification of calcium and voltage imaging

Graphs were generated in GraphPad Prism. To calculate the max ΔF/F (%) of GCaMP responses, the ΔF/F (%) of the three time points centered on the peak ΔF/F (%) after the first stimulus response were averaged. The average ΔF/F (%) of the three time points immediately preceding the stimulus onset were then subtracted to account for changing baselines during imaging. ArcLight data was similarly analyzed, except that five time points centered on the peak ΔF/F (%) and five time points prior to stimulus onset were considered.

### Statistical analysis of imaging data

Statistical analysis was performed using GraphPad Prism (GraphPad Software, La Jolla, CA). All reported values are mean ± SEM. Data were analyzed using ANOVA followed by Tukey's multiple comparisons for multiple comparisons of parametric data.

### Immunohistochemistry

To visualize GRN projections with light microscopy, males of *Gr64f-Gal4*, *Gr66a-Gal4*, *Ir94e-Gal4*, or *Ppk28-Gal4* were crossed to virgins of *UAS-Syt-HA*, *20XUAS-CsChrimson-mVenus* (attP18). Dissection and staining were carried out by FlyLight (*Gr64f-Gal4* and *Gr66a-Gal4*) or in house (*Ir94e-Gal4* and *Ppk28-Gal4*) according to the FlyLight 'IHC-Polarity Sequential Case 5' protocol (https://www.janelia.org/project-team/flylight/protocols). Samples were imaged on an LSM710 confocal microscope (Zeiss) with a Plan-Apochromat ×20/0.8 M27 objective. Images were then registered to the 2018U template using CMTK (https://www.nitrc.org/projects/cmtk) and manually segmented with VVDViewer (https://github.com/takashi310/VVD_Viewer; *Otsuna et al., 2018*; *Kawase and Rokicki, 2022*) in order to remove any nonspecific background.

### Acknowledgements

We thank Lori Horhor, Jolie Huang, Neil Ming, and Parisa Vaziri for EM tracing contributions. This work was supported by NIH R01DC013280 (KS) and NIH F32DK117671 (GS). We thank John Bogovic for registration of EM skeletons in the 2018U template. We thank Jan Funke for providing analysis of the predicted neurotransmitters used by GRNs. Neuronal reconstruction for this project took place in a collaborative CATMAID environment in which 27 labs are participating to build connectomes for specific circuits. Development and administration of the FAFB tracing environment and analysis tools were funded in part by the National Institutes of Health BRAIN Initiative grant 1RF1MH120679-01 to Davi Bock and Greg Jefferis, with software development effort and administrative support provided by Tom Kazimiers (Kazmos GmbH) and Eric Perlman (Yikes LLC). Peter Li, Viren Jain and colleagues at Google Research shared automatic segmentation (*Li et al., 2019*). Members of the Scott lab and David T Harris provided comments on the manuscript.

## Additional information

### Funding

| Funder | Grant reference number | Author |
| --- | --- | --- |
| National Institutes of Health | R01DC013280 | Kristin Scott |
| National Institutes of Health | F32DK117671 | Gabriella R Sterne |

The funders had no role in study design, data collection and interpretation, or the decision to submit the work for publication.

### Author contributions

Stefanie Engert, Conceptualization, Data curation, Formal analysis, Investigation, Methodology, Visualization, Writing - original draft, Writing - review and editing; Gabriella R Sterne, Formal analysis, Investigation, Methodology, Visualization; Davi D Bock, Resources, Supervision; Kristin Scott, Conceptualization, Funding acquisition, Project administration, Supervision, Writing - original draft, Writing - review and editing

### Author ORCIDs

Stefanie Engert http://orcid.org/0000-0003-0644-8116
Gabriella R Sterne http://orcid.org/0000-0002-7221-648X
Davi D Bock http://orcid.org/0000-0002-8218-7926
Kristin Scott http://orcid.org/0000-0003-3150-7210

### Decision letter and Author response

Decision letter https://doi.org/10.7554/eLife.78110.sa1
Author response https://doi.org/10.7554/eLife.78110.sa2

## Additional files

### Supplementary files

• Transparent reporting form

### Data availability

FAFB neuronal reconstructions are available from Virtual Fly Brain (https://fafb.catmaid.virtualflybrain.org/).

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
