## [Editor Report]

The authors reconstructed the axons of gustatory receptor neurons from the labellum in an EM volume of a whole adult *Drosophila* brain. The authors were able to correlate the EM data with light microscopic data in terms of the identity of neurons reconstructed, thus enabling the use of published functional data already available in terms of different taste modalities. This revealed that extensive synaptic connections are found between neurons of the same modality. This article will be of interest to neuroscientists working in the field of circuits and behavior, especially feeding behavior.

---

## [Decision Letter]

**Decision letter after peer review:**

Thank you for submitting your article "*Drosophila* gustatory projections are segregated by taste modality and connectivity" for consideration by *eLife*. Your article has been reviewed by 2 peer reviewers, and the evaluation has been overseen by a Reviewing Editor and K VijayRaghavan as the Senior Editor. The following individual involved in the review of your submission has agreed to reveal their identity: Stefanie Hampel (Reviewer #1).

The Reviewers agree that the paper provides new insight into morphologically distinct labellar gustatory projection subtypes and their connectivity on a synaptic level in *Drosophila*. The conclusions are well supported by data and rigorous analysis.

We would like to suggest the following revisions to improve the clarity and accessibility of the results to a general audience:

1) Improve the presentation of the results to make them visually more informative and striking. For example, an expansion of Figure 4 would be helpful whereby the dry connectivity map and diagram are integrated with a topographical/ "organotypic" and functional map. Can the projections in the SEZ (Figure 2) be correlated with the location of the sensory neurons in the labellum (based on light microscopy data), with respect to their molecular/modality/receptors?

2) Given that the authors did not reconstruct the entire GRN population it should be discussed that an additional unknown GRN class could have been missed.

*Reviewer #2 (Recommendations for the authors):*

My suggestions focus on the presentation of their data, and not any technical aspects per se. I think they could do a bit more to make their results more accessible and visually more informative and striking to others, both those working closely in the field as well as those working in more different areas. What I would love to see is an expansion of Figure 4 (either here or in a later figure as a summary) whereby the dry connectivity map and diagram are integrated with a topographical/ "organotypic" and functional map. For example, can the projections in the SEZ (Figure 2) be correlated with the location of the sensory neurons in the labellum (based on light microscopy data), with respect to their molecular/modality/receptors?

---

## [Author Response]

We would like to suggest the following revisions to improve the clarity and accessibility of the results to a general audience:1) Improve the presentation of the results to make them visually more informative and striking. For example, an expansion of Figure 4 would be helpful whereby the dry connectivity map and diagram are integrated with a topographical/ "organotypic" and functional map. Can the projections in the SEZ (Figure 2) be correlated with the location of the sensory neurons in the labellum (based on light microscopy data), with respect to their molecular/modality/receptors?

We have updated figure 4 with clearer labels to make it more informative and accessible. Unfortunately, it is not possible to correlate projections with location on the labellum. The only single neuron projection analysis that I am aware of does not map the fibers by location on the proboscis labellum (Nayak and Singh, 1985). In addition, as each chemosensory bristle contains multiple GRNs, it would be necessary to label molecularly-defined single GRNs from each bristle in order to generate such a map. Instead, we now include additional discussion on the possibility that there may be GRN subgroups based on location in the labellum or bristle subtype that further divide the groups that we categorized (ln 310-330).

2) Given that the authors did not reconstruct the entire GRN population it should be discussed that an additional unknown GRN class could have been missed.

We now include discussion that additional GRN classes or subclasses may exist (ln 307-310).

Reviewer #2 (Recommendations for the authors):My suggestions focus on the presentation of their data, and not any technical aspects per se. I think they could do a bit more to make their results more accessible and visually more informative and striking to others, both those working closely in the field as well as those working in more different areas. What I would love to see is an expansion of Figure 4 (either here or in a later figure as a summary) whereby the dry connectivity map and diagram are integrated with a topographical/ "organotypic" and functional map. For example, can the projections in the SEZ (Figure 2) be correlated with the location of the sensory neurons in the labellum (based on light microscopy data), with respect to their molecular/modality/receptors?

We appreciate this comment and have added additional information on the connectivity in figure 4 and additional labeling of the groups (by modality) to make the figure more accessible. Unfortunately, it is not possible to correlate the projections in Figure 2 based on the location in the labellum. The only single neuron projection analysis that I am aware of does not map the fibers by location on the proboscis labellum (Nayak and Singh, 1985). In addition, as each chemosensory bristle contains multiple GRNs, it would be necessary to label molecularly-defined single GRNs from each bristle in order to generate such a map. However, we have included additional discussion on the possibility that there may be GRN subgroups based on location in the labellum or bristle subtype that further divide the groups that we have categorized based on modality (ln 309-329).